# Selective Anti-Cancer Effects of Plasma-Activated Medium and Its High Efficacy with Cisplatin on Hepatocellular Carcinoma with Cancer Stem Cell Characteristics

**DOI:** 10.3390/ijms22083956

**Published:** 2021-04-12

**Authors:** Yan Li, Tianyu Tang, Hae June Lee, Kiwon Song

**Affiliations:** 1Department of Biochemistry, College of Life Science and Biotechnology, Yonsei University, Seoul 03722, Korea; liyan900506@hotmail.com; 2Department of Electrical Engineering, Pusan National University, Busan 46241, Korea; tanghutu@hotmail.com (T.T.); haejune@pusan.ac.kr (H.J.L.)

**Keywords:** cold atmospheric-pressure plasma (CAP), hepatocellular carcinoma cell lines Hep3B and Huh7 with cancer stem cell populations, plasma-activated medium (PAM), combinatorial therapy, caspase-dependent and -independent cell death

## Abstract

Hepatocellular carcinoma (HCC) is a major histological subtype of primary liver cancer. Ample evidence suggests that the pathological properties of HCC originate from hepatic cancer stem cells (CSCs), which are responsible for carcinogenesis, recurrence, and drug resistance. Cold atmospheric-pressure plasma (CAP) and plasma-activated medium (PAM) induce apoptosis in cancer cells and represent novel and powerful anti-cancer agents. This study aimed to determine the anti-cancer effect of CAP and PAM in HCC cell lines with CSC characteristics. We showed that the air-based CAP and PAM selectively induced cell death in Hep3B and Huh7 cells with CSC characteristics, but not in the normal liver cell line, MIHA. We observed both caspase-dependent and -independent cell death in the PAM-treated HCC cell lines. Moreover, we determined whether combinatorial PAM therapy with various anti-cancer agents have an additive effect on cell death in Huh7. We found that PAM highly increased the efficacy of the chemotherapeutic agent, cisplatin, while enhanced the anti-cancer effect of doxorubicin and the targeted-therapy drugs, trametinib and sorafenib to a lesser extent. These findings support the application of CAP and PAM as anti-cancer agents to induce selective cell death in cancers containing CSCs, suggesting that the combinatorial use of PAM and some specific anti-cancer agents is complemented mechanistically.

## 1. Introduction

Hepatocellular carcinoma (HCC), which accounts for 70–90% of primary liver cancers [1], is one of the five most common cancers worldwide and has a poor prognosis. High heterogeneity and recurrence rates are the major characteristics of HCC [2]. Recent studies have suggested that cancer stem cells (CSCs), a subpopulation of cancer cells, are responsible for HCC initiation, recurrence, and hierarchical organization of heterogeneous cancer cells [3]. CSCs possess stem cell-like properties and can undergo self-renewal and differentiation. It has been reported that specific subsets of CSCs confer high rates of recurrence and therapeutic resistance [4,5]. Liver CSCs can be identified using several well-established cell-surface markers such as epithelial cell adhesion molecule (EpCAM), aldehyde dehydrogenase (ALDH), and CD133 [4,6,7,8]. Moreover, the “Yamanaka-Factors,” namely, Sox2, Nanog, and Oct4, are the major transcription factors responsible for stemness. Several studies have shown that Sox2, Nanog, and Oct4 are essential for tumorigenesis and progression of various cancers [9,10,11,12]. Liver CSCs are resistant to several chemotherapeutic drugs such as doxorubicin, as well as targeted-therapy drugs such as sorafenib, thus reducing the efficacy of anti-cancer therapies [8,10,13,14]. Furthermore, the majority of the anti-cancer therapeutics have been reported to be cytotoxic to normal cells. Therefore, there is an urgent need to develop novel anti-cancer therapies that selectively target the CSCs in HCC, with minimal cytotoxicity to normal cells.

Cold atmospheric-pressure plasma (CAP) is a novel approach in cancer therapy as CAP has been shown to selectively induce apoptosis in various cancer cells [15,16,17,18,19]. CAP is a partially ionized neutral gas containing complex and highly reactive chemical species [20,21]. Typically, CAP contains a mixture of ions, electrons, free radicals, and neutral molecules, including reactive oxygen species (ROS) and reactive nitrogen species (RNS) [15]. In addition to the direct application of CAP, an indirect use involves the generation of plasma-activated medium (PAM). PAM, the culture medium irradiated with CAP, has an anti-cancer effect on various cancer cells and is as effective as CAP [22,23,24]. In addition, PAM has an advantage for applications in plasma medicine as it can be prepared in advance and stored before use [25].

Chemotherapeutic agents, such as doxorubicin and cisplatin that target DNA, and targeted-therapy drugs against specific signaling pathways, such as sorafenib and trametinib, are widely used to treat various cancers including HCC. Doxorubicin intercalates with DNA and inhibits topoisomerase II to induce DNA damage, resulting in apoptosis [26]. Another DNA damage-related alkylating agent, cisplatin, crosslinks and damages DNA to activate the DNA damage responses, subsequently inducing apoptosis in cancer cells [27,28]. Sorafenib (SFB), a tyrosine kinase inhibitor, is the first targeted-therapy drug approved for the treatment of HCC. It targets several tyrosine kinases involved in tumor progression, angiogenesis, and apoptosis and Raf kinases involved in the mitogen activated protein kinase (MAPK) cascade [14,29,30]. Trametinib is MEK1/MEK2 inhibitor, which induces apoptosis and is active against several cancers, including HCCs carrying a mutant BRAF or RAS [31]. However, these therapies are not highly effective in some HCCs and cell lines derived from them.

Evidence suggests that combinatorial treatments are more effective as they synergize to induce cell death in cancer [32]. Combinational therapies have been used to reduce drug resistance and induce synergistic cell death in cancers via specific molecular mechanisms. Previous results have also shown that the combinatorial use of CAP and diverse chemotherapies enhances cancer cell death [33,34,35]. For example, temozolomide, an alkylating agent, could enhance the killing of glioblastoma U373MG cells when combined with CAP [33]. However, the specificity and mechanisms of CAP and PAM synergy with other chemotherapies, especially in HCC, are not well defined.

In this study, we examined whether CAP and/or PAM could selectively induce cell death in HCC cells with CSC properties, while minimizing cytotoxicity in a normal hepatocyte cell line. We also studied whether the combination of PAM with different anti-cancer agents shows additive anti-cancer effects.

## 2. Results 

### 2.1. Verification of the CSC Characteristics of HCC Cell Lines 

This study aimed to determine whether CAP and PAM could selectively induce cell death in the CSC-like HCC cell lines, Hep3B and Huh7, while minimizing cytotoxicity in the normal hepatocyte cell line, MIHA. First, the CSC properties of the Hep3B and Huh7 cell lines, which contain CSC subpopulations, were verified by analyzing the expression of the cell-surface markers, CD133 and EpCAM, and compared to the immortalized normal hepatocyte cell line, MIHA. Liver CSCs can be identified using several well-established cell-surface markers such as epithelial cell adhesion molecule (EpCAM), aldehyde dehydrogenase (ALDH), and CD133 [4,6,7,8]. We found that 86.4% of Hep3B cells and 60.6% of Huh7 cells co-expressed CD133 and EpCAM, while only around 0.14% of the MIHA cells were CD133 and EpCAM positive (Figure 1A). 

To further characterize the CSC properties of Hep3B and Huh7 cells, we examined the mRNA expression of the pluripotency markers, Oct4, Sox2, and Nanog in Hep3B and Huh7 cells and compared them to MIHA cells. The Sox2 mRNA was highly expressed in Hep3B compared to MIHA cells (Figure 1B). The observations shown in Figure 1A,B demonstrated that the Hep3B and Huh7 cell populations had a subset of cells with CSC properties. 

We also examined the chemosensitivity of Hep3B and Huh7 cells to anti-cancer agents, because CSCs may contribute to their resistancy to chemotherapies. Doxorubicin is a chemotherapeutic drug used in various cancers, including HCC. The IC_50_ for doxorubicin was 0.16 μM in Hep3B, 0.14 μM in MIHA, and 0.05 μM in Huh7, demonstrating that IC_50_ of doxorubicin is a little higher in Hep3B than in MIHA, and Huh7 is more sensitive to doxorubicin than Hep3B and MIHA (Figure 1C). We also tested the chemosensitivity of these cells to sorafenib, which inhibits several tyrosine kinases in HCC [14,29,30]. IC_50_ of sorafenib was 4.08 μM in Hep3B, 3.9 μM in Huh7, and 2.01 μM in MIHA, thus, it was higher in Hep3B and Huh7 cells than MIHA (Figure 1C). These results demonstrated that Hep3B was more resistant to doxorubicin and sorafenib, compared to MIHA cells. Huh7 showed higher resistance to sorafenib than MIHA, even though it was more sensitive to doxorubicin. Taken together, these results showed that the anti-cancer drugs did not selectively induce cell death in Hep3B and Huh7 cells when compared to MIHA cells.

### 2.2. Design of a Cold Atmospheric Pressure Plasma Device Using Air as the Gas Supply

To efficiently apply CAP to the cells and generate PAM in a 35 mm culture dish, we designed a CAP-generating device to fit the culture dish. Thus, the various chemical species generated by the CAP could evenly cover the surface of the culture medium as shown in Figure 2. This device was also designed to use air as the gas supply, which has a high chance of generating radicals including various reactive nitrogen species (RNS), and does not require a separate gas tank. 

To generate atmospheric pressure plasmas (APPs) in ambient air under a gas pressure (p) of 760 Torr, we utilized a surface dielectric barrier discharge (s-DBD). A conventional DBD has two facing electrodes with a gap distance, *d*, covered by dielectrics on both or at least one electrode surface. However, in this case, the material to be treated needs to be located between the two electrodes, which makes it challenging to generate a discharge because a high voltage is required for a large value of *pd* by Paschen’s breakdown law especially with ambient air, where *p* is gas pressure and *d* is the gap distance between the two electrodes. For example, a pressure of 760 Torr and a gap distance of 1 cm requires an applied voltage larger than 20 kV. In many cases, therefore, a He or Ar noble gas flow was utilized to make a jet type plasma source for the biomedical applications of APP [36].

In our experiment, however, we utilized an s-DBD to make an air breakdown with a relatively low voltage utilizing electrode patterns onto a dielectric surface covering the other electrode. The advantage of s-DBD is that the discharge is isolated from the counterpart cell and media. In our device, an s-DBD was attached to the cover of a 35-mm culture dish to efficiently and consistently replicate the same experimental conditions as shown in Figure 2. The s-DBD has two copper electrodes on both sides of a dielectric Teflon disk with a thickness of 0.254 mm and a relative permittivity of 2.2. The surface electrode facing the cultured cells was patterned, where the plasma was generated along the electrode lines, as shown in Figure 2. The diameter of the patterned electrode was 18 mm. The electrodes were coated with 2 μm gold to increase the conductivity and to decrease contamination. A sinusoidal voltage up to 10 kV was applied with a frequency of 20 kHz from a circuit. The advantage of this device is that neither a vacuum chamber nor a gas flow is necessary. Compared to plasma jet devices using noble gases such as He and Ar [37], the breakdown voltage of the air discharge is much higher because the dissociation process dissipates the electron energy significantly for the N_2_ and O_2_ molecules. The bond-dissociation energies are 9.79 eV for N_2_ and 5.15 eV for O_2_, which are 945 and 498 kJ/mol, respectively. Especially, the triple bonding of N_2_ is too strong to dissociate, and thus, very high activation energy is required. Therefore, a high bias voltage, enough to break the bonding, should be applied. However, once it is dissociated, there is a high chance of the generation of reactive nitrogen species (RNS). Compared to the plasma jet device, the surface DBD does not generate an intense one-directional flux of radicals and ions because there is no gas flow. Thus, isotropic diffusion is the dominant mechanism for the transport of the reactive species generated. 

The advantage of this device is using ambient air without the need for any additional gas source. This was accomplished inside the closed dish, with the surface DBD attached to the bottom of the lid, as shown in Figure 2. Therefore, a closed system is sufficient to achieve the high density of plasma-generated components, including radicals, required to mediate a biological effect. 

### 2.3. CAP and PAM Showed Highly Selective Anti-Proliferative Activity in HCC Cell Lines 

We examined the relative viability of MIHA, Hep3B, and Huh7 cells exposed to CAP for 2.5 min and then further incubated for 72 h. The viability of Hep3B cells exposed to CAP was sharply reduced, compared to untreated Hep3B cells at 72 h, as shown in Figure 3A. In contrast, CAP-exposed MIHA cells showed higher viability than Hep3B cells (Figure 3A). The viability of Huh7 cells exposed to CAP was higher than Hep3B cells, but lower than that of MIHA cells (Figure 3A). These results demonstrated that CAP was effective in reducing the viability of HCCs, especially that of Hep3B cells with CSC properties compared to normal hepatocytes.

Recent studies have reported that the anti-cancer effect of PAM is as effective as CAP in various cancer cells [22,23,24]. Thus, we compared the anti-proliferative effect of CAP and CAP-exposed medium (PAM) and found that the PAM reduced the viability of MIHA, Hep3B, and Huh7 cells as efficiently as direct CAP exposure (Figure 3A). To determine the CAP-exposure time for the selective anti-proliferative effect of PAM on Hep3B and Huh7 cells but not on MIHA cells, we tried several different exposure times. We found that 2.5 min-exposed PAM efficiently decreased the viability of Hep3B and Huh7 cells, while it did not seriously reduce the viability of MIHA cells (Figure 3A). When we increased the CAP-exposure time to 3.5 min or further, the PAM decreased the viability of not only Hep3B and Huh7 cells but also MIHA cells (Appendix A). Thus, we prepared PAM with 2.5 min CAP exposure in this study.

We next asked if the anti-cancer effect of CAP and PAM depends on cell density. In the experiments shown in Figure 3A, 5 × 10^4^ cells were initially plated in 35 mm dishes to analyze the anti-proliferative effects of CAP and PAM. We then increased the number of MIHA and Hep3B cells 2-fold, and 10^5^ cells were initially seeded in 35 mm dishes to examine the effect of CAP and PAM. Hep3B was highly sensitive to CAP exposure and PAM treatment compared to MIHA cells (Appendix A). 

To further confirm the markedly reduced viability in response to CAP and PAM was due to cell death, we analyzed the CAP-exposed or the PAM-treated Hep3B, Huh7, and MIHA cells by flow cytometry, following double staining of the cells with Annexin V and PI. Annexin V is commonly used as a marker for apoptosis due to its ability to bind to membrane phospholipid phosphatidylserine, which is exposed on the outer membrane of apoptotic cells [38]. PI binding to DNA can be used to detect dead cells in which plasma membranes become permeable regardless of the mechanism of cell death [39]. The total number of Annexin V-stained cells, PI-stained cells, and Annexin V/PI double-stained cells corresponded to the dead cells. The total dead cell population was about 20% higher in CAP- or PAM-treated Hep3B cells than in the untreated cells (Figure 3B). In addition, cell death in Huh7 cells was around 10% higher in CAP- or PAM-treated cells than the untreated cells (Figure 3B). However, an increase in the dead cell population was not detected in the PAM-treated and CAP-exposed MIHA cells, as shown in Figure 3B.

These results altogether indicate that CAP and PAM have a highly selective anti-proliferative effect in HCC cells compared to MIHA cells, and PAM and CAP have similar anti-proliferative effects. Interestingly, the chemo-resistant Hep3B cells, which express high levels of stemness markers as shown in Figure 1, were highly sensitive to CAP-exposure and PAM-treatment.

### 2.4. PAM Induced Caspase-Dependent and -Independent Cell Deaths in HCC

To investigate the mechanism of the PAM-induced selective cell death in Hep3B and Huh7 cells, which have a CSC subpopulation compared to the normal hepatocyte MIHA cells, we analyzed several cell death pathways induced by PAM treatment. Apoptosis, a form of programmed cell death, is the main mechanism of cell death in response to anti-cancer drugs [40]. Caspases have a key function in the execution of apoptosis. Activation of caspase-3 by initiator caspases (caspase-8, caspase-9, or caspase-10) results in cleavage of the cellular substrate, ADP-ribose polymerase (PARP), a process which is considered a hallmark of apoptosis [41]. To evaluate the induction of apoptosis by PAM treatment, we treated Hep3B, Huh7, and MIHA cells with PAM in the presence of the pan-caspase inhibitor Z-VAD for 72 h, and then compared their viability with that of only PAM- or Z-VAD-treated cells. The caspase-3 inhibitor, Z-DEVD was also used to determine whether it can prevent the killing of Hep3B and Huh7 cells induced by PAM. There were no significant differences in cell viability between the PAM-treated cells alone and the combinatorial PAM and Z-VAD/Z-DEVD-treated cells (Figure 4A,B). We also tested the combination of 10 μM Z-VAD with PAM, since treatment with 20 μM Z-VAD alone led to low levels of toxicity in Hep3B cells (Figure 4A and Appendix A). We found that 10 μM Z-VAD did not suppress the PAM-induced killing of Hep3B as well (Appendix A).

We further examined the signaling pathways responsible for apoptosis in the PAM-treated Hep3B and Huh7 cells. Interestingly, following PAM treatment, the markers of apoptosis, cleaved-PARP and cleaved-caspase3, were detected in Hep3B cells but not in Huh7 cells (Figure 4C). These results suggested that cell death in PAM-treated Hep3B cells is caspase-dependent but other caspase-independent mechanisms might also induce cell death in the PAM-treated Hep3B and Huh7 cells.

To further investigate the mechanism of cell death in PAM-treated cells, we checked whether necroptosis, one of the caspase-independent cell death pathways, was induced in the PAM-treated Hep3B and Huh7 cells. Necroptosis or regulated necrosis, is mediated by a receptor-interacting protein kinase 1 (RIP1) and RIP3-dependent signaling pathway [42,43]. The accumulation and phosphorylation of RIP1 and RIP3 leads to increased necrosome formation and consequently to necroptosis [42,43]. However, we found that RIP3 was not expressed in the HCC cell lines, Hep3B and Huh7, while it was expressed in the colon cancer HT29 cells, which were used as a positive control for necroptosis (Figure 4D). In addition, phospho-RIP1 was not detected in the PAM-treated Hep3B and Huh7 cells. In conclusion, necroptosis did not contribute to cell death in the PAM-treated Hep3B and Huh7 cells.

Next, we determined the localization of apoptosis inducing factor (AIF), known to be an important effector of caspase-independent apoptosis, to explore the mechanism of selective cell death in HCC cell lines induced by PAM treatment. AIF translocates from the mitochondria to the cytosol as well as to the nucleus when apoptosis is induced [44]. Here, we found that AIF surrounded and localized to the nucleus in all PAM-treated Hep3B cells and partially localized to the nucleus in the PAM-treated Huh7, as observed in Figure 4E. When the percentage of cells with nuclear AIF were counted in the PAM-treated Huh7 and compared with that of untreated Huh7 cells, more than 20% of the PAM-treated cells showed nuclear localization of AIF (Figure 4F). These observations validated that caspase-independent apoptosis mediated by AIF largely contributed to cell death by PAM. However, Hep3B and Huh7 cells pre-treated with an AIF inhibitor, N-phenylmaleimide (N-P), or pre-treated both with Z-VAD and N-P did not completely suppress cell death in PAM-treated cells (Appendix A), suggesting that other cell death mechanisms might also be activated by the PAM, when cells are pre-treated with Z-VAD and/or N-P. 

LC3B processing (converting from LC3B-I to LC3B-II) as a biomarker for induction of autophagosomes [45]. We detected that LC3B II/I ratio was meaningfully increased in PAM-treated Huh7 cells comparing with the untreated control (Appendix A), suggesting that autophagic cell death might also be induced by PAM treatment in Huh7 cells.

Altogether, these results demonstrated that multiple cell death mechanisms may contribute to the cytostatic effect of PAM.

### 2.5. Selective Additive Effects of PAM and the Specific Anti-Cancer Agents in Huh7 

HCCs have been reported to be resistant to the frequently used anti-cancer agents such as doxorubicin and sorafenib [8,10,13,14]. Thus, we wanted to check the effect of these anti-cancer agents in Hep3B and Huh7 cells. As shown in Figure 1C, the IC_50_ values of doxorubicin and/or sorafenib for Hep3B and Huh7 cells were higher than that of MIHA cells. As shown in Figure 3, the viability of both Hep3B and Huh7 cells was markedly reduced by PAM treatment, but Huh7 cells were less sensitive to PAM than Hep3B cells. Thus, we asked whether the combinatorial treatment of PAM and frequently used anti-HCC drugs would have an additive effect in inducing cell death in Huh7 cells. We focused on anti-cancer drugs which induce cell death based on different molecular mechanisms. They included chemotherapeutic agents such as doxorubicin (DOX) and cisplatin (CIS), which attack DNA, and targeted-drugs against specific signaling pathways, such as sorafenib (SFB) and trametinib (TRA). To answer this question, we examined the viability of Huh7 cells after they were co-treated with PAM and each anti-cancer agent at its IC_50_. 

We first examined the viability of Huh7 cells treated with 3.9 μM sorafenib or 0.016 μM trametinib (IC_50_ of each drug for Huh7 as shown in Figure 1C and Appendix A) individually or in combination with PAM for 72 h. The combined application of sorafenib or trametinib with PAM decreased the viability further by 17% compared to treatment with sorafenib or trametinib alone (Figure 5A). We then examined the synergistic effect of the chemotherapeutic agents, doxorubicin and cisplatin, with PAM in Huh7 cells by applying 0.05 μM doxorubicin or 5.47 μM cisplatin (IC_50_ of each drug for Huh7 as shown in Figure 1C and Appendix A) individually or in combination with the PAM for 72 h. The viability of Huh7 cells was further decreased by 15% with a combination of doxorubicin with PAM (Figure 5A). Surprisingly, cisplatin showed a remarkable additive anti-cancer effect with PAM (Figure 5A). The relative cell viability of Huh7 cells treated with both cisplatin and PAM, compared to the untreated cells, was only about 27%, while the viability of Huh7 cells treated individually with cisplatin or PAM was around 50% (Figure 5A).

The combinatory effect of the anti-HCC drugs with PAM on cell death was further verified in Huh7 cells by flow cytometry after staining with Annexin V-FITC and PI. The percentage of Annexin V-stained cells, PI-stained cells, and Annexin V/PI double-stained cells were combined to quantify cell death. Consistent with the MTT assays shown in Figure 5A, the cell viability was slightly lower in the Huh7 cells co-treated with PAM and sorafenib, trametinib, or doxorubicin, compared to each single treatment (Figure 5B,C). On the other hand, the total cell death was about 17% with the combinatorial treatment of cisplatin and PAM, while in the cells treated with only PAM or cisplatin, it was about 8.4% or 9.7%, respectively (Figure 5C). Altogether, these observations demonstrated the highly additive effect of PAM with cisplatin and less combinatory effect with sorafenib, trametinib, and doxorubicin to induce cell death in Huh7 cells.

## 3. Discussion

CSCs are the main cause of resistance to various chemotherapeutic drugs including cisplatin, paclitaxel, temozolomide, etoposide, doxorubicin, and methotrexate in various cancers [46,47,48,49,50,51]. Likewise, liver CSCs in HCC are profoundly resistant to existing chemotherapeutic drugs such as doxorubicin and methotrexate, and therefore, decrease the efficacy of anti-cancer therapies in HCC [8,10,13,51]. Moreover, the anti-cancer agents display cytotoxicity when applied to normal cells, as shown in Figure 1C. Thus, it is crucial to develop new therapies that can selectively eliminate the CSCs present in HCC, while minimizing the cytotoxicity in normal cells.

Here, we showed that the CAP-exposed PAM as well as the CAP generated from a DBD-type device using air as a source of gas was very effective in selectively inducing cell death in HCC cell lines, particularly in Hep3B cells, which harbor a subpopulation of liver CSCs, while it did not affect a normal liver cell line MIHA significantly (Figure 3). These findings strongly support the potential application of CAP and PAM as new anti-cancer agents for cancers containing CSCs such as HCC (Figure 3).

How do CAP and PAM induce cell death in HCCs harboring a CSC subpopulation that is resistant to other anti-cancer therapies? We observed weak expression of markers for caspase-dependent apoptosis, cleaved caspase-3, and cleaved-PARP in the PAM-treated Hep3B cells and no expression of these markers in PAM-treated Huh7 cells (Figure 4C). In addition, we found that AIF had completely translocated to the nucleus of PAM-treated Hep3B cells and partially localized to the nucleus of PAM-treated Huh7 cells. However, the caspase inhibitors did not prevent the reduced viability induced by PAM in both Hep3B and Huh7 cells (Figure 4A,B). In addition, treatment with an AIF inhibitor (N-P) or combinatorial treatment of Z-VAD and N-P, did not suppress cell death in PAM-treated Hep3B and Huh7 cells (Appendix A). These results strongly suggest that there are other cell death pathways activated by PAM, even in the presence of Z-VAD and/or N-P. We observed the increased LC3B II/I ratio in PAM-treated Huh7, which suggest that the autophagic cell death might also be activated by PAM. However, we showed that RIP3 is not expressed in Hep3B and Huh7 cells, which is consistent with a previous report that suggests RIP3 expression is often silenced by genomic methylation near its transcriptional start site, thus, repressing RIP3-dependent necroptosis in some cancers including HCC cell lines [52]. Therefore, necroptosis does not contribute to PAM-mediated cell death in HCC cell lines. Altogether, our data strongly suggests that PAM treatment activates multiple cell death pathways, consistent with the previous reports that CAP or PAM induces cell death in several ways including caspase-dependent apoptosis, caspase-independent cell death, and autophagic cell death [17,25,33,53]. 

We observed that Hep3B cells were more sensitive to the PAM than Huh7 cells, which is consistent with our observation that both caspase-dependent and -independent apoptosis were activated in Hep3B, while AIF-related apoptosis and autophagic cell death were induced in Huh7 cells. The different responses in Hep3B and Huh7 cells suggested that the extent of cell death induced by CAP and PAM may depend on the individual physiology and signaling pathways activated in each cancer. Thus, further studies were needed to understand the mechanisms of cell death induced by CAP and PAM in various cancer cells to develop CAP and PAM as effective anti-cancer treatments.

Several reports have shown that the combinatorial use of CAP and diverse chemotherapies enhances the cell death in different cancers [33,34,35]. Conway et al. showed that temozolomide, an alkylating agent, enhances the killing of glioblastoma U373MG cells when combined with CAP [33]. In addition, Sagwal et al. demonstrated that CAP-derived oxidants in combination with anthracycline agents such as doxorubicin and epirubicin elicit effective melanoma cell death [34]. In this study, considering that Huh7 cells were less sensitive to PAM than Hep3B cells, we examined the combined effect of PAM with other frequently used anti-HCC drugs in Huh7 cells to deduce an effective approach to induce cell death in Huh7 cells. We observed a highly additive anti-proliferative effect of PAM, when combined with cisplatin, to induce cell death in Huh7 cells. Meanwhile, only slightly increased anti-proliferative effect was detected following combined treatment of PAM and another chemotherapeutic agent, doxorubicin and the targeted anti-cancer agents, sorafenib and trametinib. Then, how can we interpret the highly selective combined effect of PAM with cisplatin in Huh7 cells? Previously, a synergistic anti-cancer effect was reported using a combinatorial treatment of anthracyclines such as doxorubicin with alkylating agents such as cisplatin in many cancers including advanced endometrial carcinoma, suggesting that these two anti-cancer drugs induce cell death via different mechanisms [54,55]. For example, inhibition of DNA topoisomerase II alpha by doxorubicin may interrupt the efficient repair of DNA damage caused by cisplatin in EMT6 mouse mammary carcinoma or human glioblastoma cells [56,57]. Thus, PAM treatment may block the repair of DNA damage induced by cisplatin or prevent the development of resistance to cisplatin, which might explain the great combined effect of PAM and cisplatin.

Previous reports have shown that even though the mechanism of action of sorafenib and trametinib involve the MAPK pathway, their combinatorial use has a synergistic effect in patients with advanced HCC [58]. On the other hand, the combined application of sorafenib with doxorubicin or cisplatin in Hep3B or in patients with advanced HCC was not effective [59,60,61,62]. These reports suggest that the efficacy of the combinatorial treatment of various anti-cancer therapies is hard to estimate, since it depends on diverse parameters including the associated therapeutic and resistance mechanisms, as well as the genetic background of each cancer cell. Thus, a combination therapy involving CAP or PAM with other anti-cancer drugs need further study to evaluate their efficiency and molecular mechanisms, prior to their clinical application in the future.

CAP primarily generates short-lived species such as hydroxyl radicals (OH), nitric oxide radicals (NO), superoxide radicals (O_2_^−^), atomic oxygen (O), singlet oxygen (^1^O_2_), and excited nitrogen (N) [15,63]. CAP-generated short-lived RONS are converted to H_2_O_2_ and NO derivatives, NO_2_^−^ and NO_3_^−^, in the PAM [63]. The RONS are known as the main factors to induce cell death in cancers [19,24,33,64,65,66]. However, several other results have shown that cell death induced by CAP and PAM are ROS independent [33,64]. Thus, the components and mechanisms of anti-cancer effect of CAP and PAM remain controversial. Studies are needed to detect the component(s) of CAP and PAM, which are responsible for the selective anti-proliferative effect in HCC cells to develop CAP and PAM for clinical applications.

## 4. Materials and Methods

### 4.1. Cell Culture

The HCC cell lines, Hep3B (obtained from the Korean Cell Line Bank, Seoul, Korea), and Huh7 (kindly provided by Professor Young Nyun Park, College of Medicine, Yonsei University, Seoul, Korea), and the immortalized normal hepatocyte cell line, MIHA (kindly provided by Professor Suk Woo Nam, Catholic University, Seoul, Korea) were maintained in high glucose Dulbecco’s modified Eagle’s medium (DMEM; Gibco, New York, NY, USA) including sodium pyruvate with 10% (*v*/*v*) fetal bovine serum (FBS; Gibco) and 1% antibiotic-antimycotic (Gibco) at 37 °C in 5% CO_2_.

### 4.2. Quantitative Real-Time PCR

MIHA, Hep3B, and Huh7 cells were seeded in 60 mm dishes at 3 × 10^5^ cells/dish, incubated for 48 h, and then harvested. Total RNA was extracted from the cells using a RNeasy Mini Kit (Qiagen, Hilden, Germany) and 400 ng RNA was used for cDNA synthesis. One ng/μL cDNA was synthesized using a Biotechnology Power cDNA Synthesis Kit (Takara Bio Inc., Shiga, Japan). Quantitative real-time PCR was performed using the TB Green^®^ Premix Ex Taq™ (Takara Bio Inc., Shiga, Japan) and QuantStudio3 real-time PCR system (Thermo Fisher Scientific, Waltham, MA, USA) using the primers listed in Table 1. The relative expression of each gene was normalized to β-actin. The normalized fold-change of mRNA levels in MIHA, Hep3B, and Huh7 cells was determined using the 2^−ΔΔCT^ method.

### 4.3. Cell Exposure to CAP and PAM

We utilized an s-DBD device to generate plasma by the breakdown of air using a relatively low voltage as shown in Figure 2. A voltage of 4.7 kV and a discharge power of 0.87 W were used. MIHA, Hep3B, and Huh7 cells were seeded in 35 mm dishes at 5 × 10^4^ cells/dish, incubated for 18 h, and then exposed to CAP for 2.5 min. For PAM treatment, 1.5 mL of DMEM in each 35 mm dish was exposed to CAP for 2.5 min and then added to the cells. Following CAP exposure or PAM treatment, the cells were incubated for a further 72 h and cell viability was detected using the 3-(4,5-dimethylthiazol-2-yl)-2,5-diphenyltetrazolium bromide (MTT) assay. 

### 4.4. Cell Viability Assay

Cell viability was analyzed using the MTT assay (Sigma-Aldrich, St. Louis, MO, USA). Cells treated with 1 mL of 0.5 mg/mL MTT solution were incubated for 1.5 h and the resulting formazan was dissolved in 1 mL dimethyl sulfoxide (DMSO). Absorbance was measured at 570 nm using a microplate reader (SoftMax Pro 4.0, Molecular Devices, San Jose, CA, USA).

### 4.5. Flow Cytometry Analysis

To detect the expression of cell surface markers, MIHA, Hep3B, and Huh7 cells were seeded in 60 mm dishes at 3 × 10^5^ cells/dish, incubated for 48 h, and then harvested. The cells were washed and stained with anti-EpCAM-FITC (1:50, BD Biosciences, San Jose, CA, USA) and anti-CD133-APC (1:50, Miltenyi, Bergisch Gladbach, Germany) antibodies for 10 min in the dark at 4 °C according to the manufacturer’s protocol. Unstained cells served as the gating control for each cell line. 

For cell death analysis, untreated, CAP-exposed, or PAM-treated, MIHA, Hep3B, and Huh7 cells were harvested 72 h after the initial treatment. The cells were washed with phosphate buffer saline (PBS), resuspended in 1X binding buffer, followed by incubation with Annexin V-fluorescein isothiocyanate (Annexin V-FITC) and propidium iodide (PI; BD Biosciences) for 15 min according to the manufacturer’s protocol. 

After staining, 10,000 cells were analyzed per assay using a FACSCalibur (BD Biosciences) flow cytometer and FlowJo V10 software. 

### 4.6. Preparation of Cell Lysates and Western Blot Analysis

Cells were harvested and lysed as previously described [67]. Cell lysates were separated by 7.5% or 12.5% sodium dodecyl sulfate polyacrylamide gel electrophoresis (SDS-PAGE) and transferred to polyvinylidene difluoride (PVDF) membranes (Merck Millipore, Burlington, MA, USA). To detect the protein of interest, anti-PARP (1:3000), anti-caspase-3 (1:1000), anti-cleaved caspase-3 (1:1000), anti-histone H3 (1:3000), anti-Phospho-RIP1 (S166) (1:1000), anti-RIP3 (1:1000), anti-β-actin (1:3000) (all from Cell Signaling Technology, Danvers, MA, USA), Phospho-RIP3 (Abcam, Cambridge, UK), and RIP1 (BD Biosciences) were used as primary antibodies. HRP-conjugated goat anti-mouse and anti-rabbit IgG (Santa Cruz Biotechnology, Dallas, TX, USA) were used as secondary antibodies. 

### 4.7. Determination of IC_50_ of Drugs, and the Combinatorial Treatment of Each Drug with PAM

The half-maximal inhibitory concentration (IC_50_) is defined as the concentration of a drug necessary to inhibit the biological activity of the target or pathway by 50%. We determined the IC_50_ of doxorubicin and sorafenib (both from Cayman Chemical Company, Ann Arbor, MA, USA) in MIHA, Hep3B, and Huh7 cells by treating cells with the indicated concentrations of doxorubicin or sorafenib (Figure 1). The IC_50_ of trametinib and cisplatin (both from Cayman Chemical Company) in Huh7 cells was also determined to examine their combinatorial effects with PAM. Further, 5 × 10^4^ cells seeded in 35 mm dishes were incubated for 18 h before treatment with anti-cancer drugs. Cell viability was measured 72 h after drug treatment using an MTT assay. 

### 4.8. Fluorescence Microscopy

Hep3B and Huh7 cells were seeded onto collagen-coated coverslips (25 mm) in 35 mm dishes, fixed in 100% methanol, and then permeabilized with 0.5% Triton X-100. After permeabilization, the cells were blocked with 3% BSA for 1 h at room temperature, followed by immunostaining for AIF and mounted in 4′,6-diamidino-2-phenylindole (DAPI) containing medium (VECTASHIELD^®^ MOUNTING MEDIUM with DAPI, Vector Laboratories, Burlingame, CA, USA). An anti-AIF antibody (1:50; Santa Cruz Biotechnology) was used as the primary antibody and Alexa Fluor 488 goat anti-rabbit IgG (Invitrogen) was used as the secondary antibody. The cells were visualized on a Zeiss Axioplan2 fluorescence microscope (Carl Zeiss, Gina, Germany) and images were acquired using an Axiocam CCD (Carl Zeiss) camera and AxioVision software (Carl Zeiss).

### 4.9. Treatment of Caspase-Inhibitor 

The pan-caspase inhibitor Z-VAD and the specific caspase-3 inhibitor Z-DEVD (both from R&D systems, Minneapolis, MN, USA) were dissolved in DMSO. Prior to PAM treatment, MIHA, Hep3B, and Huh7 cells were pretreated with 20 μM Z-VAD for 1 h, and Hep3B and Huh7 cells were treated with 50 μM Z-DEVD as indicated. The treated cells were incubated further for 72 h and cell viability was measured using an MTT assay.

### 4.10. Statistical Analyses

All data from at least three independent experiments were expressed as mean ± standard deviation (SD). Statistical analysis was performed using Student’s *t*-test on the GraphPad Prism software (GraphPad Software Inc., San Diego, CA, USA). * *p* < 0.05 was considered statistically significant.

## 5. Conclusions

In this study, we have demonstrated that air-based CAP effectively induced cell death in HCC cell lines containing CSC subpopulations, Hep3B and Huh7, especially in Hep3B cells, which shows greater chemo-resistance. In contrast, CAP exposure to MIHA, a normal liver cell line, resulted in only a marginal decrease in viability. Moreover, PAM was as efficient as CAP in inducing cell death in Hep3B and Huh7 cells, while it was not very cytotoxic to MIHA cells. These efficient and selective anti-proliferative effects of the CAP and PAM strongly support their use as novel therapeutic agents for HCCs with CSC subpopulations.

We also examined efficacy of the combined application of PAM and various anti-cancer drugs to deduce their synergistic effects in Huh7 cells. We found that the combined treatment of PAM and cisplatin was highly additive in inducing cell death in Huh7 cells, while the combination of PAM with sorafenib, doxorubicin, or trametinib was less effective. These results suggested that the synergistic effect of PAM and anti-cancer drugs is very specific, and not all PAM and anti-cancer agent combinations are mechanistically complemented.

To study the mechanism of cell death induced by the PAM in Hep3B and Huh7 cells, we performed the PAM treatment in the presence of caspase inhibitors, Z-VAD and Z-DEVD, and observed that these inhibitors did not suppress the reduced viability of Hep3B and Huh7 cells mediated by the PAM treatment. A weak expression of cleaved caspase-3 was only detected in the PAM-treated Hep3B cells, but not in the Huh7 cells. We also observed AIF activation in PAM-treated Hep3B and Huh7 cells. Taken together, these observations strongly suggest that the PAM treatment, unlike other anti-cancer drugs, does not activate a single cell death pathway but likely triggers multiple cell death pathways in Hep3B and Huh7 cells. 

## Figures and Tables

**Figure 1 ijms-22-03956-f001:**
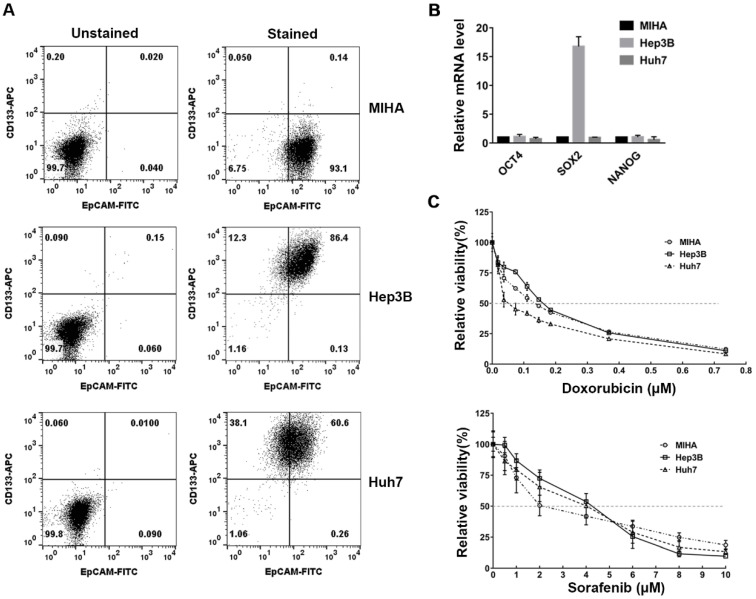
Analysis of cancer stem cell (CSC) markers and the sensitivity to several anti-cancer drugs in hepatocellular carcinoma (HCC) cell lines. (**A**) Flow cytometry was performed to analyze the expression of the cell surface markers CD133. and epithelial cell adhesion molecule (EpCAM), in Hep3B, Huh7, and MIHA cells. Unstained cells were used as the gating control for each cell line. (**B**) The mRNA expression of Oct4, Sox2, and Nanog in Hep3B, Huh7, and MIHA cells was analyzed by quantitative real-time PCR. The relative gene expression was normalized to β-actin. The results are presented as the mean ± standard deviation (SD) from three independent experiments. (**C**) Doxorubicin and sorafenib dissolved in dimethyl sulfoxide (DMSO) were diluted to indicated concentrations with DMEM. Hep3B, Huh7, and MIHA cells, seeded in 35 mm dishes, were pre-incubated for 18 h and treated with various concentrations of doxorubicin and sorafenib for 72 h. The viability was analyzed using a 3-(4,5-dimethylthiazol-2-yl)-2,5-diphenyltetrazolium bromide (MTT) assay and the relative viability was calculated as the ratio of the viability of treated to untreated cells at 72 h. The relative viability is presented as the mean ± SD of three independent experiments.

**Figure 2 ijms-22-03956-f002:**
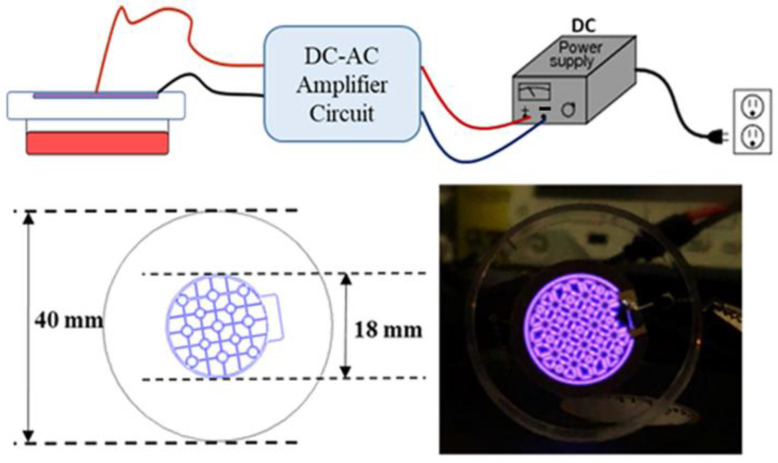
Design of an air-based cold atmospheric-pressure plasma (CAP) device used in this study. A schematic diagram of the dielectric barrier discharge (DBD) type device using air as a gas source for CAP generation is shown in this study. The CAP generated by this device evenly covers the surface of the culture medium.

**Figure 3 ijms-22-03956-f003:**
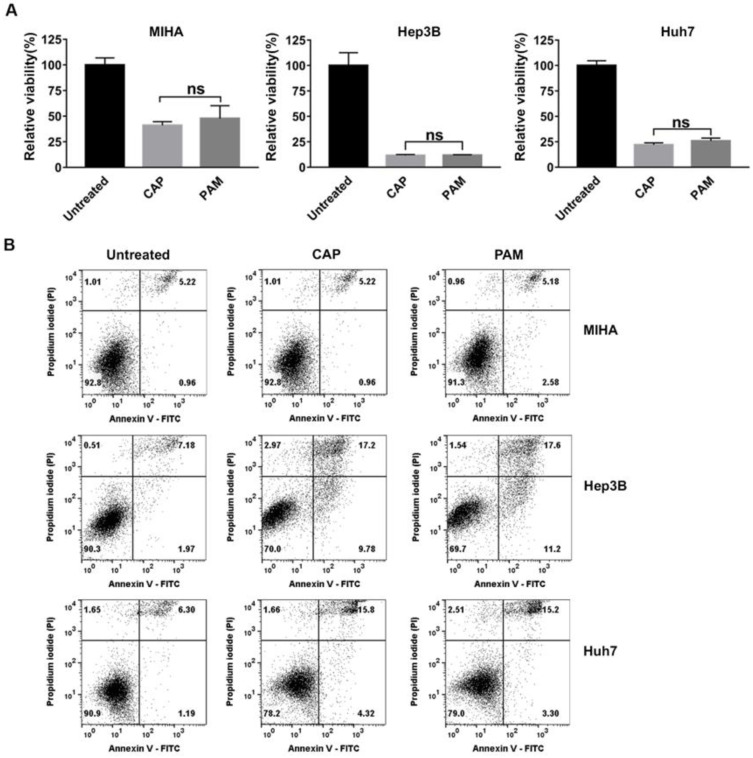
CAP and plasma-activated medium (PAM) showed a highly selective anti-proliferative effect in HCC cells. (**A**,**B**) For this, 5 × 10^4^ MIHA, Hep3B, and Huh7 cells were seeded in 35 mm culture dishes and pre-incubated for 18 h. (**A**) The cells were exposed to CAP for 2.5 min or treated with CAP-treated medium, PAM. Cell viability was analyzed at 72 h using an MTT assay. The relative viability was calculated as the ratio of the viability of treated to untreated cells at 72 h. The results are plotted as the mean ± SD of three independent experiments. ns indicates not significant (*p* > 0.05). (**B**) Cell death was examined by flow cytometry in cells analyzed in (**A**), after staining the cells with Annexin V-fluorescein isothiocyanate (Annexin V-FITC) and propidium iodide (PI). Untreated cells were used as a negative control.

**Figure 4 ijms-22-03956-f004:**
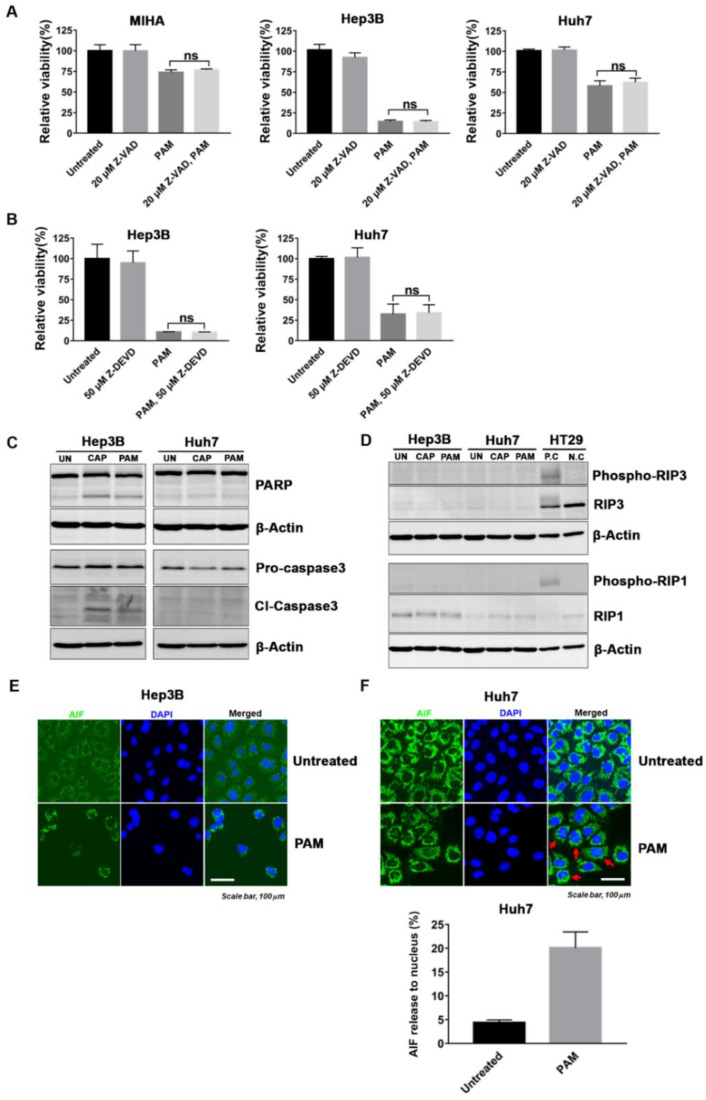
PAM induced caspase-dependent and -independent cell death in HCC. (**A**) MIHA, Hep3B, and Huh7 cells were pre-treated with the pan-caspase inhibitor 20 μM Z-VAD for 1 h followed by PAM treatment. (**B**) Hep3B and Huh7 cells were treated with the caspase-3 inhibitor 50 μM Z-DEVD and PAM. (**A**,**B**) These cells were incubated further for 72 h, and cell viability was measured by an MTT assay. Data are shown as mean ± SD of at least three independent experiments. ns indicates not significant (*p* > 0.05). (**C**–**F**) After PAM treatment (PAM), Hep3B and Huh7 cells were incubated for a further 72 h. Untreated (Un) cells were used as a negative control. (**C**,**D**) Western blot analyses for (**C**) apoptosis and (**D**) necroptosis. The expression of cleaved-caspase3 (Cl-caspase3), cleaved-ADP-ribose polymerase (Cl-PARP), Phospho-RIP3, RIP3, Phospho-RIP1, and RIP1 are shown. β-actin was used as a loading control. The colon cancer cell line, HT29, processed for necroptosis was used as a positive control (P. C.) and not processed for necroptosis was used as a negative control (N. C.). (**E**,**F**) The localization of apoptosis inducing factor (AIF) was monitored by fluorescence microscopy in Hep3B and Huh7 cells that were treated with the PAM or left untreated, and stained with AIF antibody and 4′,6-diamidino-2-phenylindole (DAPI). Red arrow indicates the nuclear localization of AIF in Huh7 cells. (**F**) The percentage (%) of cells with AIF localized in the nucleus was quantified and expressed as PAM-treated Huh7 compared to the untreated Huh7. The results are presented as the mean ± SD of three independent experiments.

**Figure 5 ijms-22-03956-f005:**
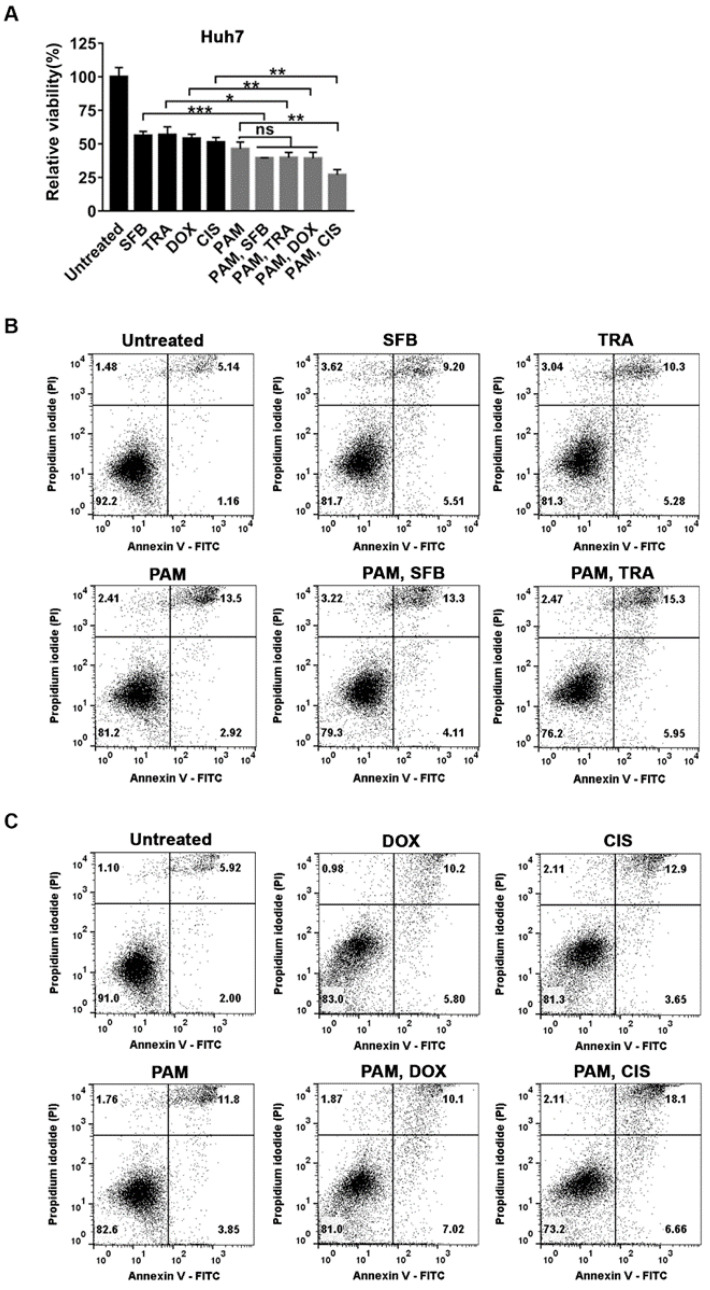
Combinatorial application of PAM and various anti-cancer agents in Huh7 cells. Analysis of the effect of PAM and the IC_50_ dose of each anti-cancer agent individually or in combination on the viability of Huh7 cells. (**A**) The viability of cells at 72 h after each treatment was measured using an MTT assay. The relative viability was calculated as the ratio of the viability of the treated to the untreated cells at 72 h. The results are plotted as mean ± SD of at least three independent experiments. * *p* < 0.05, ** *p* < 0.01, and *** *p* < 0.001 indicate significant difference. ns, not significant. (**B**,**C**) Cell death was examined by flow cytometry after staining the cells with Annexin V-FITC and PI. Untreated cells were used as negative controls.

**Table 1 ijms-22-03956-t001:** Primer sequences used for quantitative real-time PCR.

Gene		Primer Sequences
*Oct4*	Forward	5′-GACAACAATGAAAATCTTCAGGAGA-3′
	Reverse	5′-CTGGCGCCGGTTACAGAACCA-3′
*Nanog*	Forward	5′-AGTCCCAAAGGCAAACAACCCACTTC-3′
	Reverse	5′-TGCTGGAGGCTGAGGTATTTCTGTCTC-3′
*Sox2*	Forward	5′-GAGCTTTGCAGGAAGTTTGC-3′
	Reverse	5′-GCAAGAAGCCTCTCCTTGAA-3′
*β-Actin*	Forward	5′-TCCCTGGAGAAGAGCTACGA-3′
	Reverse	5′-AGCACTGTGTTGGCGTACAG-3′

## Data Availability

Not applicable.

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
