# Peer review of "Selective Anti-Cancer Effects of Plasma-Activated Medium and Its High Efficacy with Cisplatin on Hepatocellular Carcinoma with Cancer Stem Cell Characteristics"

_ijms, 2021, doi:10.3390/ijms22083956_

Round 1

Reviewer 1 Report

In this manuscript, the authors demonstrated that PAM effectively induced apoptosis in HCC cell lines through the caspase-dependent and -independent cell death pathways. Besides, they suggested the possible application of CAP and PAM for the treatment of HCCs with CSC subpopulations. This is an attractive study, however, a reviewer has a comment that should be clarified.

The authors showed the anti-tumor effects of PAM on HCC cell lines with CSC subpopulations. Which components of PAM are responsible for its anti-tumor effects on HCC? Did the authors check the components of PAM in this study? The detailed information regarding PAM are missing in this study. This should be clarified and discussed in this manuscript.

Author Response

As the reviewer pointed out, we also understood the importance of investigating the components of PAM responsible for its selective anti-tumor effects on HCC cell lines. Thus, we studied the anti-proliferative effects of the reactive oxygen and nitrogen species (RONS) of PAM, which have been reported to induce cell death in cancer cells by other groups. To our surprise, we found that the PAM we used (which contains pyruvate as a hydrogen peroxide scavenger in the medium) was able to  induce cell death independently of hydrogen peroxide, singlet oxygen, nitric oxide, and nitrite/nitrate. Hydrogen peroxide of the PAM induced cell death when there was no hydrogen peroxide scavenger (pyruvate) in the medium, but its effect was not selective on HCC cell lines. Because the study of the components of PAM for the selective anti-proliferative effect on HCC cell lines became lengthy, we prepared it as a separate manuscript. This manuscript also is ready to be submitted, and I attach the pdf file of this manuscript for your reference. We are willing to submit this manuscript to IJMS to be published as a back-to-back paper with the paper on revision, if you suggest so.

We briefly added the paragraph about the putative functional components of PAM to induce cell death and their arguing points in the Discussion of the revision.

Reviewer 2 Report

This paper by Y. Li et al brings new important results concerning direct or indirect low temperature atmospheric pressure plasma cancer treatment and is clearly of high interest. Beside confirming the plasma treatment (both direct and indirect)  selectivity between cancer cells and normal cells, it shows the efficient combined effect of associated chemotherapy and PAM (plasma activated medium) treatment on hepatocellular carcinoma with cancer stem cell characteristics for various anti-cancer agents with a particularly remarkable effect in the case of the use of cisplatin. Materials and methods are well described and perfectly adapted to the present study. I recommend the publication of this paper with minor revisions. I have only the following comments:

- Title and lines 317 to 327

Is “the synergistic effect” the appropriate wording? Synergistic means that the result obtained with the combination of two actions is better than the sum of the expected results of the individual actions. This is not at all the case for dorafenib, trametinib and doxorubicin (the wording of line 326 “milder synergy” is not appropriate). In the case of cisplatin, the result obtained with the combined treatment corresponds approximately to an addition of the effect of the two separated treatments (50% relative viability due to PAM adds to the 50% relative viability due to cisplatin which leads around  25% (27% measured) relative viability total, which is great and not expected!!!, but in my opinion is not “synergistic”. A synergistic effect would have been to obtain a relative cell viability below 25 % taking into account the individual effect of each treatment. Comments from the authors on that point, and changes in the title and in the text accordingly would be very welcome.

Author Response

By following the review’s comment, we removed “the synergistic effect” from the title and the text, and switched to “the additive effect” in the text if necessary.

Reviewer 3 Report

In this manuscript, Li et al. study the effects of cold atmospheric pressure plasma (CAP) and plasma-activated medium (PAM) in HCC cells with cancer stem cell-like characteristics. Specifically, the authors observed that both CAP and PAM induced cell death in HCC cells with PAM triggering both caspase- dependent and independent cell death mechanisms. In addition, they also showed the selective enhanced efficacy of cisplatin in combination with PAM, suggesting the use of combinatorial use of PAM and other anti-cancer drugs in HCC. The study is straightforward and the data is sound and well presented. However, I have several clarifications that needs to be addressed as follows:

  • Page 3, 2.1: The authors should explain their rationale for choosing CD133/EpCAM as their CSC marker of choice and why not others like CD90, CD13, CD44 etc?
  • Page 3, 2.1: The authors looked at the chemosensitivity of Hep3B and Huh7 to doxorubicin and sorafenib and showed the differential chemosensitivity of the cell lines towards the drugs. This differential chemosensitivity could be due to the intrinsic resistance contributed by other factors rather than CSC property as the authors did not sort out the CSC population (i.e. CD133/EpCAM+) and compare their drug sensitivity against their non-CSC counterparts (i.e. CD133/EpCAM-). This would provide a stronger argument for CSC contributing to drug resistance.
  • The authors investigated the cell death mechanisms induced by PAM treatment. They showed differential cell death responses in 2 HCC cell lines induced by PAM, highlighting the involvement of both caspase-dependent and -independent pathways. However, the authors should also explore the possibility of autophagy as there are reports demonstrating the PAM-induction of autophagy in other cancers. In addition, did the authors perform a time- or dose-dependent PAM treatment on both cell lines? The lack of cell death responses in Huh7 might be due to the lack of effective PAM dose.
  • Page 10-11, 2.5: The authors attributed the enhanced reduction in cell viability and increased Annexin V (apoptosis) due to the synergistic effect of cisplatin and PAM combination. However, the authors did not perform drug-dose synergy experiments, hence should be careful with the usage of “synergy” in their main text. In addition, Annexin V is a probe used for identifying and quantifying apoptotic cells, however in Fig 4C, the authors concluded that PAM-treatment in Huh7 is via caspase-independent pathway. There is a contradiction in results between the 2 experiments that the authors should carefully reanalyse.
  • There is also a lack of discussion about PAM-induced ROS or RNS in their discussion. The authors should avoid reiterating their results extensively in the discussion and focus on how their current work compared with existing literature.

Minor comments:

Figure 1 caption “Analysis of CSC markers in HCC cell lines” should include drug treatment as well.

Page 4, Figure 1C: why 2 different concentration units (µg/ml and µM) were used? Is it possible to use 1 concentration unit for easy comparison and consistency throughout the manuscript.

Page 9, Figure 4E: The AIF image for Hep3B is overexposed with high green fluorescence background. Please use another image with appropriate exposure. “bar” should be “scale bar”

Author Response

Page 3, 2.1: The authors should explain their rationale for choosing CD133/EpCAM as their CSC marker of choice and why not others like CD90, CD13, CD44 etc?

As the reviewer pointed out, various cell surface markers have been reported to be associated with stemness and have been used to identify CSC subpopulations. We used CD133 and EpCAM as markers for Hep3B and Huh7 cells because they are most well-known and widely used markers for CSC in HCC (Yamashita et al., 2009, Gastroenterology; Chen et al., 2011, Journal of Hepatology). Other markers such as CD90 (0% in MIHA, 0.47% in Hep3B, 1.15% in Huh7) and CD44 (0.15% in MIHA, 0.17% in Hep3B, 2.03% in Huh7) are expressed at very low levels (Yang et al., 2008, Cancer cell). In case of CD90, Yamashita et al. even showed that no CD90 expression was detected in Hep3B and Huh7 cells by immunofluorescence and FACS analysis. Haraguchi et al. demonstrated CD13 as a candidate liver cancer stem cell marker, but the subpopulation of CD13 is only about 3.1% in Huh7 (Yamashita et al., 2016, International Journal of Oncology) and a low level (not described) in Hep3B cells. (Haraguchi et al., 2010, The Journal of Clinical Investigation). We only focused on CSC markers with relatively high levels of expressions in Hep3B and Huh7 cells. As we mentioned in the manuscript, 86.4% of Hep3B cells and 60.6% of Huh7 cells co-expressed CD133 and EpCAM in our experiments.

Page 3, 2.1: The authors looked at the chemosensitivity of Hep3B and Huh7 to doxorubicin and sorafenib and showed the differential chemosensitivity of the cell lines towards the drugs. This differential chemosensitivity could be due to the intrinsic resistance contributed by other factors rather than CSC property as the authors did not sort out the CSC population (i.e. CD133/EpCAM+) and compare their drug sensitivity against their non-CSC counterparts (i.e. CD133/EpCAM-). This would provide a stronger argument for CSC contributing to drug resistance.

As the reviewer pointed out, we could not be certain whether the differential chemo-sensitivity of Hep3B and Huh7 is due to other intrinsic factors or the CSC populations. Thus, we soften the sentence in Page 5 line 108. In fact, we tried to sort out the CSC population (CD133/EpCAM+) and non-CSC (CD133/EpCAM-) cells but failed due to technical problems.

The authors investigated the cell death mechanisms induced by PAM treatment. They showed differential cell death responses in 2 HCC cell lines induced by PAM, highlighting the involvement of both caspase-dependent and -independent pathways. However, the authors should also explore the possibility of autophagy as there are reports demonstrating the PAM-induction of autophagy in other cancers.

We also considered autophagic cell death by PAM, as the reviewer pointed out. We detected that the LC3II/LCI ratio was fairly increased in the Huh7 cells treated with PAM comparing with the untreated control, as shown below. This result suggests that autophagic cell death is also induced by PAM at least in Huh7 cells that did not show any evidence of caspase-dependent cell death by PAM. We added this result as Fig. S4 and mentioned in the Results and Discussion of the revised manuscript.

In addition, did the authors perform a time- or dose-dependent PAM treatment on both cell lines? The lack of cell death responses in Huh7 might be due to the lack of effective PAM dose.

As the reviewer asked, we checked the effect of PAM in a CAP exposure time-dependent manner. To determine the CAP-exposure time for the selective anti-proliferative effect of PAM on Hep3B and Huh7 cells but not on MIHA cells, we tried several different exposure times. We found that 2.5 min-exposed PAM efficiently decreased the viability of Hep3B and Huh7 cells, while it did not seriously reduce the viability of MIHA cells (Fig. 3A). When we increased the CAP-exposure time to 3.5 min and 4 min to prepare PAM, these exposure times decreased the viability of not only Hep3B and Huh7 cells but MIHA cells as well (the result was added as Fig. S1 in the revised manuscript.). Thus, we prepared PAM with 2.5 min CAP exposure in this study. We added these explanations in the revised manuscript lines 189-195.

Page 10-11, 2.5: The authors attributed the enhanced reduction in cell viability and increased Annexin V (apoptosis) due to the synergistic effect of cisplatin and PAM combination. However, the authors did not perform drug-dose synergy experiments, hence should be careful with the usage of “synergy” in their main text.

By following the review’s comment, we removed “the synergistic effect” from the title and the text, and switched to “the additive effect” in the text if necessary.

In addition, Annexin V is a probe used for identifying and quantifying apoptotic cells, however in Fig 4C, the authors concluded that PAM-treatment in Huh7 is via caspase-independent pathway. There is a contradiction in results between the 2 experiments that the authors should carefully reanalyse.

In the FACS analysis of PAM-treated Hep3B and Huh7 cells, we observed that PAM-treated Huh7 cells mainly showed annexin V and PI double-stained populations, while ~10% of only annexin V-stained cells were present in Hep3B (Fig. 3B). The cells only stained with annexin V are likely the cells in the early apoptotic stage by caspase activation. However, PI-strained cells represent dead cells. We could not detect the Huh7 cells only stained by annexin V, and consistently the caspase activation was not recognized in Huh7 cells (Fig. 4C). We detected the AIF-related cell death (caspase-independent cell death) in PAM-treated Huh7 cells. AIF-related cell death also entails annexin V staining by phosphatidylserine exposure (Reviewed in Sevrioukova, 2011 and Cregan et al., 2004). Thus, we thought that the annexin V-positive Huh7 cells could be accounted by the caspase-independent AIF-related cell death.

There is also a lack of discussion about PAM-induced ROS or RNS in their discussion. The authors should avoid reiterating their results extensively in the discussion and focus on how their current work compared with existing literature.

As the reviewer pointed out, we also understood the importance of investigating the components of PAM responsible for its selective anti-tumor effects on HCC cell lines. Thus, we studied the anti-proliferative effects of the reactive oxygen and nitrogen species (RONS) of PAM, which have been reported to induce cell death in cancer cells by other groups. To our surprise, we found that the PAM we used (which contains pyruvate as a hydrogen peroxide scavenger in the medium) can induce cell death independently of hydrogen peroxide, singlet oxygen, nitric oxide, and nitrite/nitrate. Hydrogen peroxide of the PAM induced cell death when there was no hydrogen peroxide scavenger (pyruvate) in the medium, but its effect was not selective on HCC cell lines. Because the study of the components of PAM for the selective anti-proliferative effect on HCC cell lines became lengthy, we prepared it as a separate manuscript. This manuscript also is ready to be submitted, and I attach the pdf file of this manuscript for your reference. We are willing to submit this manuscript to IJMS to be published as a back-to-back paper with the paper on revision, if you suggest so.

We briefly added the paragraph about the putative functional components of PAM to induce cell death and their arguing points in the Discussion.

Minor comments:

Figure 1 caption “Analysis of CSC markers in HCC cell lines” should include drug treatment as well.

By following the comment of the reviewer, we changed the caption of Fig. 1 as “Analysis of CSC markers and the sensitivity to several anti-cancer drugs in HCC cell lines” in the revised manuscript.

Page 4, Figure 1C: why 2 different concentration units (µg/ml and µM) were used? Is it possible to use 1 concentration unit for easy comparison and consistency throughout the manuscript?

As the reviewer suggested, we used the same concentration unit (μM) for all anti-cancer drugs used in the revised manuscript.

Page 9, Figure 4E: The AIF image for Hep3B is overexposed with high green fluorescence background. Please use another image with appropriate exposure. “bar” should be “scale bar”

As suggested by the reviewer, the AIF image for Hep3B in Figure 4E was switched to an image with less green fluorescence background in the revision. “bar” is changed to “scale bar”.

Round 2

Reviewer 1 Report

The authors adequately revised their manuscript according to the reviewers comments. Therefore, this manuscript is suitable for publication in the journal of “International Journal of Molecular Sciences”.